# Aging of Wood for Musical Instruments: Analysis of Changes in Color, Surface Morphology, Chemical, and Physical-Acoustical Properties during UV and Thermal Exposure

**DOI:** 10.3390/polym15071794

**Published:** 2023-04-05

**Authors:** Lidia Gurău, Maria Cristina Timar, Camelia Coșereanu, Mihaela Cosnita, Mariana Domnica Stanciu

**Affiliations:** 1Faculty of Furniture Design and Wood Engineering, Transilvania University of Brasov, 500036 Brasov, Romania; 2Department of Product Design Mechatronics and Environment, Transilvania University of Brasov, 29 Eroilor, 500036 Brasov, Romania; 3Faculty of Mechanical Engineering, Transilvania University of Brașov, B-dul Eroilor 29, 500360 Brasov, Romania

**Keywords:** resonance wood, acoustic quality, sound propagation, UV aging, color changes, chemical changes, surface morphology

## Abstract

The acoustic features of old resonance wood in violins exhibit a superior quality when compared to those from new resonance wood. This study focuses on an assessment of the sound quality of two types of wood for musical instruments, spruce and maple (class A and D), before and after aging via thermal and UV exposure. The samples were characterized before and after UV aging in terms of color change (using a Chroma meter), surface morphology (using a MarSurf XT20 instrument), chemical changes (monitored by FTIR spectroscopy), and sound propagation speed (using an ultrasound device). After UV treatment, the wavier surface increased the area of exposure and degradation. Also, the color changes were found to be more accentuated in the case of spruce compared to sycamore maple. The FTIR results indicated more advanced aging processes for spruce when compared to maple under the same experimental conditions. This difference resulted mostly from the increased formation of carbonyl-containing chromophores via oxidative processes in spruce rather than in maple, which is in agreement with the color change findings. Exposure of both species to thermal and UV radiation led to an increase in sound propagation speed, both longitudinally and radially, and to a greater extent in wood quality class A when compared to quality class D.

## 1. Introduction

The acoustic quality of old violins has aroused the interest of researchers to explore the causes that lead to higher sound quality when comparing geometrically identical violins that have been manufactured from recently made resonance wood. According to some researchers, the natural aging of wood is responsible for increasing the acoustic quality of violins. Thus, ref. [1,2] highlight the occurrence of significant differences in the wood density of old violins compared to the wood of new violins. The decrease in density by 33% in spruce boards and 10% in maple boards caused changes in the stiffness distribution, which can have a direct impact on vibration effectiveness or can indirectly change sound radiation due to the modified damping characteristics. On the other hand, Su et al. 2021 [3] examined the effects of natural aging on Cremonese spruce using C13 solid-state nuclear magnetic resonance, noting that there was little sign of degradation, which was in contrast to Cremonese sycamore wood that showed significant hemicellulose degradation and the demethoxylation of lignin. Such distinctions may be caused by fundamental differences in the chemical compositions of wood. However, aging can also mean the formation of new substances and stabilization, which can, in some cases, be a desirable effect [4]. The critical wavelengths to break the most important chemical bonds in wood polymers are 346, 334, and 289 nm. These wavelengths are found in the UV component of solar radiation and affect carbon-carbon, carbon-oxygen, and carbon-hydrogen bonds, respectively [5].

In addition to photodegradation, surface thermal degradation must also be considered because the surface temperature of outdoor irradiated wood can reach 60–90 °C, depending on its color [6]. Wood aging also leads to changes in the size of micropores, considering that extended aging time makes the wood microstructure more stable, resulting in a decrease in the number of micropores [7]. All investigations into naturally or artificially aged wood have highlighted both chemical and physical changes, with these being connected to each other. Thus, the color changes of wood as a result of natural aging are related to specific chemical changes involving both the main and secondary chemical components of wood [8,9] that are associated with the modification of other physical properties, such as surface roughness [10,11] or wetting properties and adhesion capacity [12,13]. Changes in chemical and mechanical properties are not limited to the wood surface. Mechanical properties can also be affected by aging wood under variable conditions of temperature and relative humidity [14,15,16,17]. It is certain that the aging process of wood differs depending on the environmental conditions (outdoors or indoors). When wood is exposed to direct sunlight, severe damage is induced by UV radiation [9,10,18,19]. In the situation of objects stored indoors, as is the case of stringed musical instruments, the sunlight spectrum is mostly filtered by window panes, resulting in a decrease in the intensity of UV radiation. In addition, in indoor environments, artificial light sources are also present. Due to different light spectrums, the photodegradation processes of indoor wood are not the same as those for outdoor wood. The studies carried out by artificial aging procedures on different wood species have shown that the color changes are rapid in the first 12 h of exposure for hardwoods and for the first 24 h for softwoods [9,20,21,22]. A continuation of the exposure time beyond these limits has shown a decrease in the fading speed. Most wood species recorded almost no further color change until the end of the irradiation period. 

Several studies have addressed the problem of the photodegradation of wooden surfaces due to UV and thermal radiation, but very few of them have analyzed the acoustic and physical changes produced in the resonance wood used for musical instruments [17,23,24,25,26,27]. Noguch et al. 2012 [17] and Obataya 2017 [24] observed an increase in the stiffness of naturally aged wood by approximately 38% in 121-year-old wood, after which a decrease was observed with increasing aging time. The speed of sound was 14% higher for the 276-year-old wood compared to the 8-year-old wood. Similar results were also obtained by Kranitz et al. [16], who observed an increase in the dynamic modulus of elasticity in the longitudinal direction with tree age.

The main objective of this work is to study whether the effect of artificial aging through thermal and UV exposure can improve the acoustic parameters of two wood species that are commonly used for musical instruments: spruce and sycamore maple. The novelty of the paper consists of a holistic research approach, in which the aging effect is regarded not only upon acoustics solely but in combination with other changes, such as on surface color, surface morphology, and chemical modifications, in order to provide a better understanding of the aging process. 

## 2. Materials and Methods

### 2.1. Materials

The wood samples selected for this study belong to the two most used wood species for the construction of violins: spruce (*Picea abies* (L.) *Karst.*) and sycamore maple (*Acer pseudoplatanus*, L.). Two quality class samples were prepared, depending on their anatomical characteristics: class A and class D, selected according to the anatomical classification from previous studies [28,29]. Spruce wood samples were coded MA (class A) and MD (class D), respectively, and differed in the number and width of the annual rings, with the finest structure attributed to class A. Sycamore maple wood samples were coded PA (class A with a curly grain) and PD (class D with a straight grain). The sample code used after exposure to ultraviolet radiation was the UV suffix, for example, MAUV (spruce, quality A, and exposure to UV). The wood was harvested from the forests of the Carpathian Mountains. It was cut and naturally dried for 3 years (class D) and 10 years (class A) to obtain the semi-finished products necessary for the manufacturing of stringed musical instruments. The samples were processed in the form of plates with dimensions of 240 mm × 80 mm × 4 mm, respecting the main directions of wood (length (longitudinal direction) × width (radial direction) × thickness (tangential direction)). All plates were sanded with 220-grit sandpaper. Three replicates were studied for each sample category, as shown in Figure 1.

### 2.2. Methods

#### 2.2.1. UV and Thermal Exposure

The samples were introduced into a photocatalytic reactor ICD – Transilvania University of Brașov, Brașov, Romania, equipped with 2 UV tubes (Philips, TL-D BLB 18 W/108) and 5 VIS light tubes (Philips, TL-D Super 80 18 W/865). Each tube emits radiation in the UV range, with wavelengths between 340–400 nm and wavelength λmax (emission) = 365 nm; the average value of UV irradiation was measured with a Delta-T pyranometer, type BF3, with 3 W/m^2^ [30]. Wood samples were exposed to UV for 360 h [31]. The distance between the samples and the lamp during irradiation was 20 cm, and the temperature inside the irradiation chamber was 50 °C. Physical and acoustic properties were periodically evaluated and measured. The moisture content was monitored using a Merlin HM8-WS1 moisture meter (Merlin Technology Inc. 1441 E. Business Center Dr., Mount Prospect, Illinois, USA), and the mass was measured with a Kern and Sohn Weighing Balance, EWJ 600-2SM (Mechanikus Gottlieb KERN, Balingen, Germany).

#### 2.2.2. Wood Color Change Measurements 

The degree of change in the wood color as a result of the progressive exposure to UV and to thermal radiation was determined by monitoring the color parameters with a chroma meter CR-400 Konica Minolta. Thus, the (L*) lightness values were obtained, as well as the chromatic co-ordinates on the green-red axes (the degree of green/red), a*, the degree of blue/yellow, b*, and the total color change, denoted $\Delta E_{ab}^{*}$. The latter was calculated with Equation (1), according to [22,32,33]:(1)$$\Delta E_{ab}^{*}=\sqrt{{\left(\Delta L^{*}\right)}^{2}+{\left(\Delta a^{*}\right)}^{2}+{\left(\Delta b^{*}\right)}^{2}}$$
where Δ*L*^∗^—the lightness difference, Δ*a*^∗^—the redness difference, and Δ*b*^∗^—the yellowness difference between wood before and after exposure to UV radiation.

The color was measured in three points on the plate, which were positioned according to the schematic representation in Figure 2. The guide marks served to maintain the same location when monitoring the color before and during UV exposure.

#### 2.2.3. Surface Quality Measurements

The equipment used for the evaluation of the surface morphology comprised a MarSurf XT20 instrument manufactured by MAHR Gottingen GMBH (Göttingen, Germany), fitted with an MFW 250 scanning head, with a tracing arm in the range of ±750 µm, and a stylus with a 2 µm tip radius and a 90° tip angle. The stylus moved at a speed of 0.5 mm/s and exerted a low scanning force of 0.7 mN at a lateral resolution of 5 µm. Three measurements per specimen were recorded by following an identical trace, before and after exposure to UV, according to the measurement principles from Figure 3. The scanned profiles were processed with MARWIN XR20 software provided by the instrument supplier (Göttingen, Germany). The measured profiles contain a superposition of deviations from the nominal flat surface, surface waviness, and surface roughness [34]. By removing the form deviations, a primary profile was obtained, which contained the waviness and roughness together. A special filtering procedure using a cut-off value separated the roughness profile from the primary profile. A robust Gaussian regression filter with a cut-off length of 2.5 mm was used in this research [34]. Mean value parameters were calculated, such as Ra (arithmetical mean deviation of the roughness profile), Rt (total height of profile), and Wa (arithmetical mean deviation of the waviness profile) (from ISO 4287:2009) [35]. Other parameters included the Abbot-curve parameters: Rk (the core roughness depth), Rpk (the reduced peak height), and Rvk (the reduced valley depth) from ISO 13565-2:1998 [36], explained in detail in [37]. The parameter Rk is closely related to “processing roughness” or “core roughness” [37], being the most stable parameter (small standard deviation) and the least influenced by the presence of isolated protrusions or valleys of the profile. Rpk is useful in assessing the surface fuzziness in a profile, while Rvk is sensitive to surface deep cavities occurring below the core roughness.

#### 2.2.4. FTIR Analysis of Chemical Changes on Wood Surfaces

The chemical changes produced by UV and thermal exposure were investigated using an FTIR spectrometer (ALPHA, Bruker, Ettlingen, Germany) with an ATR (attenuated total reflectance) module. The spectra were recorded in the range 4000–600 cm^−1^ at a resolution of 4 cm^−1^ and 24 scans per sample. Spectra were recorded in 3 areas for each sample. The recorded spectra were further processed for baseline correction and smoothing, and average spectra were computed for each category of sample/treatment using the OPUS software (version 7.2, Bruker, Ettlingen, Germany). Average spectra were further normalized (min-max normalization) and compared to highlight any chemical changes due to aging. The working method was adopted according to [9,10].

Furthermore, the normalized average spectra were integrated, and the ratios of the areas of some relevant absorption bands of the main chemical components of wood were calculated to better highlight the chemical changes brought about by aging. The selected absorption bands were those at around 1506–1507 cm^−1^, assigned to aromatic skeletal vibration of lignin, 1730–1740 cm^−1^, assigned to unconjugated carbonyl groups, and about 1370 cm^−1^, assigned to holocellulose and considered as an internal reference [38]. The calculated ratios were: A1506/A1370, A1730/A1370, and A1730/A1506, as in previously reported research [9].

#### 2.2.5. Sound Velocity Measurements

The measurement of the sound propagation speed in wood in the longitudinal (VLL) and radial (VRR) direction was carried out using a LUCCHIMETTER portable ultrasound device, Cremona, Italy. The principle of the ultrasound (US) measurements is presented in Figure 4 [17]. The density of each plate was also determined in each stage of the experiment, and the longitudinal and radial modulus of elasticity was calculated in accordance with the calculation relationship mentioned by [26]. The mean (MV) and standard deviation (SD) values of the measured parameters were recorded before and during exposure to temperature and UV. The rate of change of Young’s modulus with exposure time was calculated as a percentage difference in the values measured at successive exposure durations. 

## 3. Results and Discussions

### 3.1. Changes in Color Parameters

Table 1 presents the mean values and the standard deviations of the L*, a*, and b* co-ordinates for all studied samples before and after UV exposure. The ΔE* and ΔL* total variations were also calculated and recorded. The color changes differed between the two species—spruce and sycamore maple—but the differences were minor between the two quality classes A and D within the same species, as can be seen in Figure 5 and in the color differences from Table 1. Exposure to UV caused a change in brightness (by decreasing the value of this parameter), with the speed of change reaching the maximum value in the first 72 h of exposure. Thus, after the first 72 h of exposure to the UV radiation, the wood surface darkened by 3.43% in the PA samples, 4.76% in the PD samples, 4.36% in the MA spruce samples, and 4.35% in the MD samples. 

The spruce darkened more compared to the sycamore maple samples, recording a total color variation of around 15 when compared to the sycamore maple, whose total color variation was around 10, as can be seen in Table 1. The results obtained are in agreement with those reported by [21,39,40], where the change in color for spruce was the largest among all the investigated species. Some researchers [10] found that the sycamore maple that was subjected to a long simulated natural aging process indoors suffered from a dramatic decrease in brightness after 4–5 months of exposure. After this time interval, the process was reversed so that the degree of brightness partially recovered (approximately 45%) after 12 months of natural aging simulation [10]. When sycamore maple was exposed for longer (84 months), the color darkened again. It is worth mentioning that time 0 from the graphs in Figure 5 represents the values of the parameters measured before exposure to UV in the initial stage. Figure 6 shows the difference in color between the initial samples and the accelerated aging ones, at the end of the exposure.

### 3.2. Study on the Surface Morphology

#### 3.2.1. Surface Morphology Parameters for Spruce and Maple, before and after the UV Treatment

Among the surface parameters evaluated, Table 2 contains the ones which proved to be the most sensitive to UV exposure. They will be analyzed in the following subsections in combination with the measured profiles. The Rk parameter is a measure of the central area of the profile, where most of the profile data are concentrated, and is the most indicated parameter for quality assessment, being an expression of the majority of measured points on a surface [34]. The Rk mean values before and after the UV treatment are depicted for both species and quality classes in Figure 7.

#### 3.2.2. The Effect of UV Treatment on Spruce Wood

Since the same measuring trace was followed before and after the UV treatment, this allowed not only for a reliable comparison between the roughness parameters but also for a visual comparison of the profiles. A first observation was that the spruce samples bulged after the UV treatment, which can be seen in Figure 8, showing a measured spruce profile (MA quality) before and after the UV treatment. The green arrow indicates an accentuation of the deviation from flatness of approx. 100 microns in amplitude. Another observation was that the sanded spruce surfaces contain a specific wavy aspect which is related to the sequence of earlywood–latewood, and their frequency depends on the frequency of the annual rings (number of occurrences on a given length). This seems to be a typical reaction of spruce wood after sanding. Deep gaps occur at the beginning of a new annual ring, corresponding to the first earlywood tissue; then, the wood forms a bump covering the earlywood–latewood sequence until the next annual growth border, where a new gap is formed. This “anatomical waviness” is accentuated by the UV treatment, and this behavior can be observed by the naked eye and by touch, as well as by the evidence in Figure 9, where the primary profiles of spruce MA and spruce MD are shown. For example, the surface waviness, expressed by Wa, increased more than three times in the case of MA spruce treated with UV (Table 2). This phenomenon was also observed by [41,42] in the case of the photodegradation of wood combined with thermal oxidation.

The anatomical waviness of the MA spruce class, characterized by narrower annual rings, gave birth to a higher frequency anatomical waviness when compared to the spruce MD class, where the annual rings were wider (Figure 9). 

The separation of roughness irregularities from waviness is made by selecting a filter cut-off. The value that is recommended for wood surfaces, namely 2.5 mm, in accordance with [34], can retain the high-frequency waviness (anatomical waviness) in the roughness profile, which was mainly the case for the spruce MA class (Figure 10) and less for MD. An increase in magnitude for the anatomical waviness is clearly visible (the blue profile) after UV exposure.

A wavier surface means an increased area of exposure to UV and temperature modification and, thus, a higher likelihood for surface chemical degradation, which will be discussed in Section 3.3. Retaining the anatomical waviness in the roughness profile had an impact on the assessment of the roughness of the MA quality spruce. The central roughness (Rk) of the profile and also the average roughness (Ra) were higher after UV treatment by approx. 22% (Table 2, Figure 7). In the case of the MD quality spruce, the roughness changes produced by the UV treatment were negligible, approx. 1% in the case of Rk and 2% in the case of Ra. Even if the anatomical waviness increased slightly after exposure to UV radiation, the impact on the roughness was reduced.

#### 3.2.3. The Effect of UV Treatment on the Surface Morphology of Sycamore Maple Wood

As in the case of spruce, the exposure of maple to UV light caused surface bulging (Figure 11), and this behavior was present for both qualities of wood. Another observation was that the UV treatment accentuated the surface waviness of the curly maple, where the Wa increased by app. three times (Table 2). As in the case of spruce, an increase in surface waviness causes more area exposure to the modification agent. From Table 2 and Figure 7, it can be observed that before and also after UV treatment, the core roughness (Rk) of the curly maple was smaller than that of the straight-grain maple. The slightly higher density of curly maple (659 kg/m^3^) in comparison to straight grain maple (626 kg/m^3^), together with different local hardnesses caused by the variation in grain orientation in the curly maple (from along to almost across the grain), might have been responsible for a smoother sanded surface for the curly maple. The effect of the UV treatment on maple wood can be seen in Table 2 and Figure 7 via the discrete decrease in the central roughness of the profile, Rk, by approx. 6–7%, which may be a consequence of a shrinkage phenomenon, narrowing the initial wood cavities and so reducing the ability of the stylus to penetrate. 

The profile comparisons in Figure 12a for curly maple show a discrete shrinkage phenomenon occurring after the treatment, where some of the features have slightly changed their initial location. Another observation in the comparative profiles of Figure 12 was that some of the deepest features (possible earlywood pores) deepened further, which was perhaps due to a cracking effect under treatment. This caused an increase in the Rvk parameter by approx. 11% in PA quality and approx. 9% in PD quality (Table 2).

It can be concluded that the UV treatment has accentuated the shape deviations from a flat surface and has enlarged the area exposure by accentuating the surface waviness. This phenomenon was more pronounced for quality class A in both species. The surface roughness evaluated by the parameter Rk has a tendency to increase after exposure to UV in the case of spruce samples regardless of the class, and this is mainly attributed to an increase in high-frequency anatomical waviness being retained in the roughness profiles. A slight decrease in the core roughness (Rk) for sycamore maple wood may be attributed to surface shrinkage.

### 3.3. FTIR Spectroscopy

The comparative spectra for spruce and sycamore maple (Figure 13), as well as the data in Table 3, illustrate the common chemical characteristics of the two wood species as well as some of the differentiating features, such as an increased absorption intensity at 1730 cm^−1^, which is assigned to unconjugated carbonyl groups that were observed for sycamore maple wood when compared to spruce wood. This can be associated with a higher content of acetyl groups that are mainly present in hemicelluloses of a pentosan type (e.g., xylan), which are found in higher percentages in hardwoods compared to softwoods [43,44]. 

The FTIR analysis also highlighted differences in the characteristic absorptions of lignins; for spruce, an absorption at 1261 cm^−1^, which is assigned to guaiacyl (G) ring breathing (characteristic for the guaiacyl type lignin (GL) in softwoods), was identified, while for the maple samples, well-defined absorptions at 1325 cm^−1^ and 1235 cm^−1^, which are assigned to syringyl (S) rings and C-O stretching in lignin and xylan (with contributions from the syringil ring, respectively), were identified due to the different nature of hardwood lignins, which are of a siryngil-guaiacyl nature (SGL). Additionally, the absorption of the aromatic skeleton vibration of lignin was recorded at 1507 cm^−1^ for spruce and at 1504 cm^−1^ for sycamore maple due to the same structural differences [9,38,44,45,46]. On the other hand, the absorption of the aromatic nucleus at approximately 1593 cm^−1^ appeared only as a shoulder in the spruce wood spectrum, whereas it was clearly distinct in the maple wood spectrum.

The calculated ratios of the areas of the relevant absorption peaks A1730/A1370 (unconjugated carbonyls/holocellulose) and A1504–1507/A1370 (lignin/holocellulose) (centralized in Table 3) highlighted the differences between the two studied species. Thus, the calculated ratios A1730/A1370 for sycamore maple wood in its initial state (before UV treatment) were 3.91–3.93, which is higher than those calculated for spruce wood: 1.51–1.87 due to a larger amount of acetyl groups in the hemicelluloses (mostly pentosans) that were reported in the global holocellulose content (cellulose and hemicelluloses). On the other hand, the calculated ratios A1504–A1507/A1370 of 1.96–1.99 for spruce compared to 1.11–1.37 for maple are in accordance with the higher content of lignin reported in holocellulose for softwoods compared to hardwoods [43]. 

The comparative spectra recorded for the sycamore maple and spruce wood before and after UV exposure indicated chemical changes that affected lignin, which is the main UV absorber in wood structures. A decrease in the most characteristic absorption band of lignin (1504–1507 cm^−1^) was clearly observed, as well as the intensification of the absorption band at 1730 cm^−1^, which is characteristic of nonconjugated carbonyl groups. These changes highlight the degradation of lignin and the formation of chromophore com-pounds with carbonyl groups (Figure 14a,b), which is in good accordance with the literature [8,9,10,12,38,45,46,47]. 

The results of the semi-quantitative evaluation of the chemical change after UV exposure via the calculated absolute and relative ratios of the selected relevant absorptions are contained in Table 3. 

The relative ratios were calculated by reporting the absolute ratios after UV exposure to their initial values, allowing for an overview of the processes occurring during the experimental aging procedure and a comparative evaluation of the behavior of the two species. Thus, the decrease in the ratios between lignin and holocellulose (A1510/A1370), resulting in relative ratios lower than 1.0 (0.75–0.69 for spruce; 0.82–0.61 for sycamore maple) highlighted the degradation of lignin regardless of species and quality class, while the increase in the ratios of nonconjugated carbonyl to holocellulose (A1730/A1370), resulting in relative ratios higher than 1.0 (2.09–2.36 for spruce; 1.19–1.28 for sycamore maple) highlighted oxidative processes, with the formation of carbonyl-containing chromophores. 

Based on this relative semi-quantitative evaluation, lignin degradation was almost similar for the two species under study, with some variations for both species depending on the quality class of the wood. Lignin degradation appeared to be slightly more intense for the samples in quality class A, both in the case of spruce (8.67%) and sycamore maple (34%), but no conclusion should be drawn based only on this type of evaluation. 

The relative ratios of A1730/A1370 clearly suggest more advanced oxidative processes in the case of the spruce wood samples than in the case of maple wood. The generation of carbonyl groups as a result of lignin photo-oxidation was found to be much higher in softwoods when compared to hardwoods by other researchers [38,45]. Based on the relative FTIR ratios calculated in this research, some small differences appeared between the quality classes for both wood species. For the spruce wood class A samples, the formation of carbonyl chromophores seemed to be 13% lower than for the D class samples, while for the sycamore maple class A samples, the oxidative process seemed more intense, at approximately 7.8% higher than for the class D samples. The two chemical processes occurring in parallel during the UV-induced photo-oxidation of wood (lignin degradation and the oxidation of lignin degradation products, forming colored compounds containing carbonyl chromophores) are cumulatively illustrated by the increase in the ratios of nonconjugated carbonyl/lignin (A1730/A1510), with their relative values higher than 1.0 after UV exposure (Table 3). Values of 2.78–3.41 for the spruce samples (classes A and D) and 1.56–1.64 for the sycamore maple samples (classes A and D) indicate more advanced aging processes for spruce compared to sycamore maple under the experimental conditions employed in this research. This difference resulted mostly from the higher formation of carbonyl-containing chromophores via oxidative processes, which is in good accordance with the greater color changes measured in the spruce samples (ΔE = 14.74–15.17) when compared to sycamore maple samples (ΔE = 9.26–10.50) presented in Table 1.

### 3.4. Changes in Acoustic Properties

After thermal and UV aging, all the wood samples used for the construction of musical instruments registered a decrease in density; approximately 19% for the spruce wood samples and approximately 10% for the maple samples (Figure 15a,b). Exposure to UV radiation had the effect of increasing the speed of sound propagation in the wood, both longitudinally and radially, in both species. It was found that the changes produced by the lignin degradation processes (highlighted in Section 3.3.) led to an increase in the sound propagation speed by 5–6% in the longitudinal direction in both the spruce wood and sycamore maple wood. In the radial direction, the increase in the sound speed reached up to 15%, with the anatomical quality class being an indicator of differentiation. Thus, in the case of the quality class A samples, the sound velocity in the radial direction increased after the UV treatment by 14.5% for spruce and 15.5% for sycamore maple when compared to the values obtained for the class D samples (9.7% spruce; 11.5% maple) (Figure 15c–f). The Young’s modulus values in the longitudinal direction, for which the calculation is based on the product of the speed squared and the density, recorded an increase after exposure to UV radiation, with 3% for the PA samples, 6% for the PD samples, 3.1% for the MA samples, and 3.5% for the MD samples. The most notable changes were registered in the radial direction, where the modulus of elasticity had the largest increase in the class D samples for both spruce and maple (Figure 15g–j).

Table 4 summarizes the acoustic and elastic parameters measured during exposure to thermal and UV treatment. It can be observed that the only property that registered a decrease in value was the density of both species and qualities. 

The other properties (sound velocity, as well as the modulus of elasticity in the longitudinal and radial direction) increased after repeated exposure to UV. Their highest increase was obtained in the radial direction regardless of the species. From a qualitative point of view, it can be concluded that the UV and thermal treatment led to an improvement in the acoustic and elastic parameters of the wood selected for musical instruments, as characterized by the relatively low thickness (maximum 4 mm) studied in this paper. In agreement with the above findings, the literature data [17,24,25,28,48,49,50,51,52,53] confirm that the ultrasound velocity and dynamic elastic modulus of the aged wood shift to higher values when compared to that of recently produced wood. 

In Figure 16, the stiffness rate of change for all samples is represented, which is calculated as a percentage difference between the values at successive exposure durations. It can be observed that the samples from class A, both spruce and sycamore maple, show the same trend for the elasticity modulus in both the longitudinal and radial directions. The most important changes occur at the beginning of the exposure (after 72 h), and then a decrease in the rate of change is recorded. After 300 h, in the case of the sycamore maple samples, the rate of change of the analyzed parameter increases again. In the class D samples, the most important changes in the longitudinal direction occur from 200 h of aging onwards. 

## 4. Conclusions

The aim of this paper was to analyze the changes induced via exposure to UV radiation and temperature on the physical, chemical, and acoustic properties of two quality classes (A and D) of spruce and sycamore maple wood. The main results are summarized as follows:The UV treatment induced an increase in the total color change, which was more accentuated in spruce compared to sycamore maple. However, the differences in the color change between the two quality classes (A and D) were insignificant;The FTIR results indicated a more advanced aging process for spruce compared to maple under the same experimental conditions. This difference resulted mostly from the formation of more carbonyl-containing chromophores by oxidative processes, which is in good accordance with the greater color changes measured in the spruce samples compared to the sycamore maple samples;A wavier surface after the UV treatment increased the area of exposure and degradation, and this was more pronounced for quality class A;For both species, the sound speed increased after the UV treatment in thee longitudinal as well as in the radial direction. UV exposure improved the sound speed to the greatest extent for quality class A (of the two species) and for radial direction sound propagation. The quality class A samples registered a sound speed increase in the radial direction of 14.5% (spruce) and 15.5% (maple) when compared to class D (9.7% spruce; 11.5% sycamore maple). The sound velocity increased by app. 5–6% in the longitudinal direction;The most important changes in the stiffness of the class A boards (spruce and sycamore maple) occurred in the first 72 h of exposure to thermal and UV radiation and after 200 h for the class D samples.

## Figures and Tables

**Figure 1 polymers-15-01794-f001:**
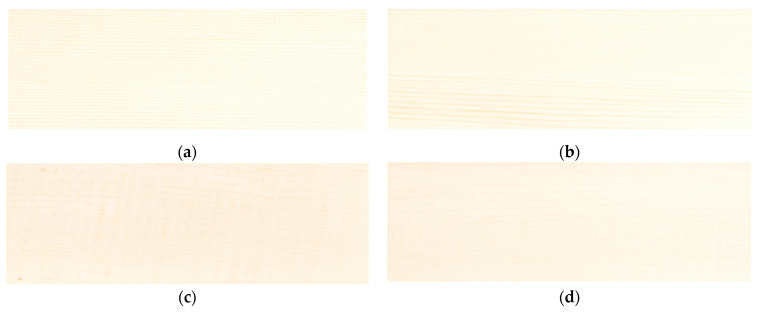
The types of samples studied: (**a**) spruce samples, grade A; (**b**) spruce samples, grade D; (**c**) sycamore maple samples, grade A; (**d**) sycamore maple samples, grade D.

**Figure 2 polymers-15-01794-f002:**
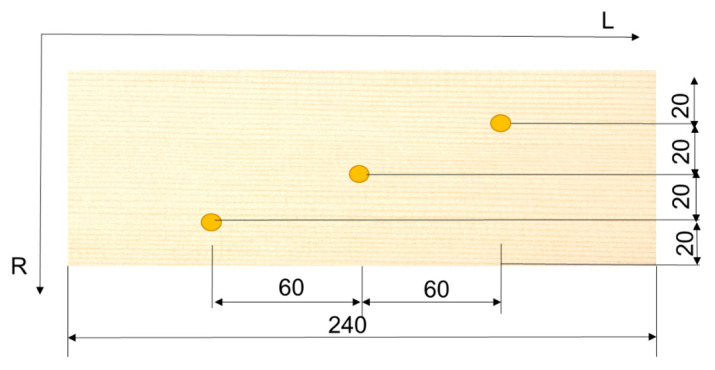
The color measurement location.

**Figure 3 polymers-15-01794-f003:**
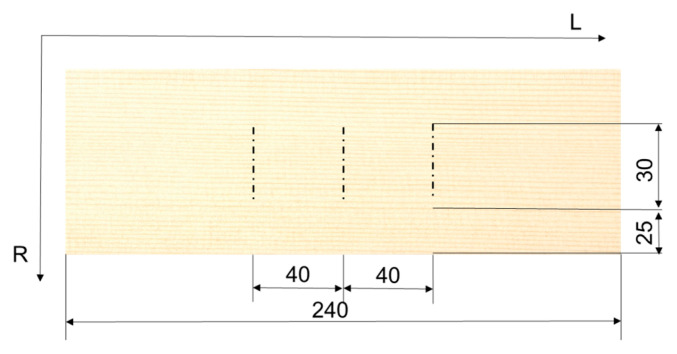
The measuring principle for the surface morphology of the samples. Values in mm.

**Figure 4 polymers-15-01794-f004:**
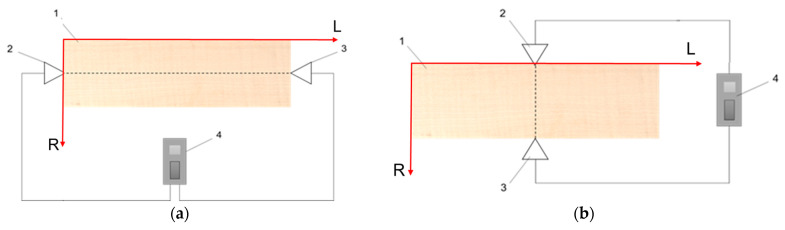
Measuring the sound velocity in wood: (**a**) in the longitudinal direction; (**b**) in the radial direction. Legend: 1–sample; 2–transmitter; 3–receiver; 4–US transmitter; acquisition and displayed data device.

**Figure 5 polymers-15-01794-f005:**
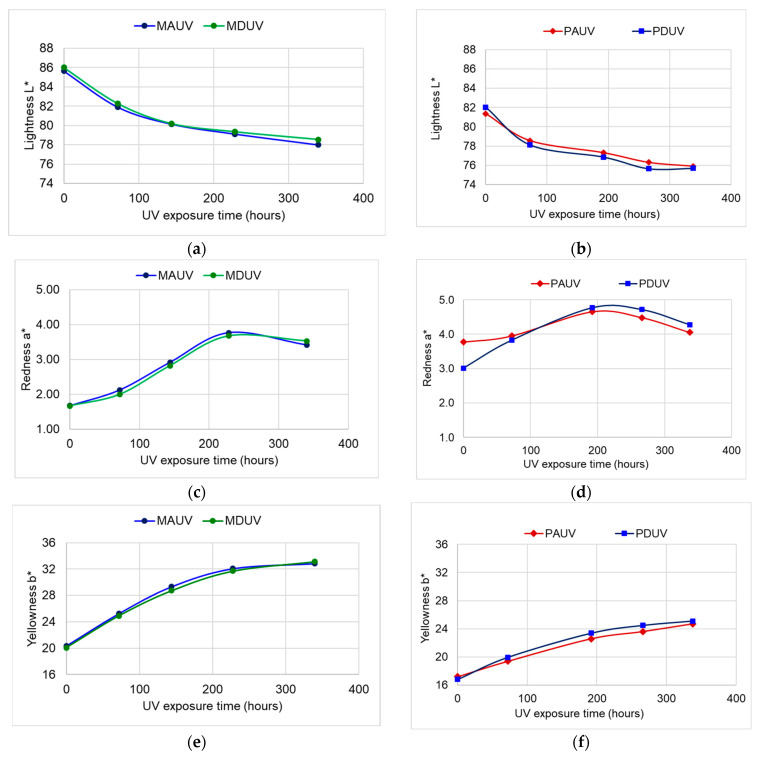
The variation of color parameters with an increase in UV exposure time: (**a**) lightness of spruce samples; (**b**) lightness of sycamore maple samples; (**c**) redness of spruce samples; (**d**) redness of sycamore maple samples; (**e**) yellowness of spruce samples; (**f**) yellowness of sycamore maple samples.

**Figure 6 polymers-15-01794-f006:**
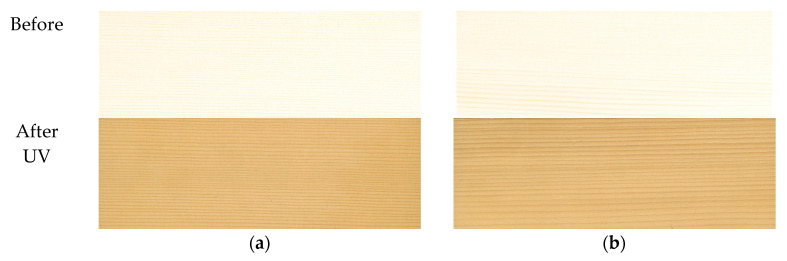
Color comparison between wood samples before and after artificial aging: (**a**) spruce samples, grade A; (**b**) spruce samples, grade D; (**c**) sycamore maple samples, grade A; (**d**) sycamore maple samples, grade D.

**Figure 7 polymers-15-01794-f007:**
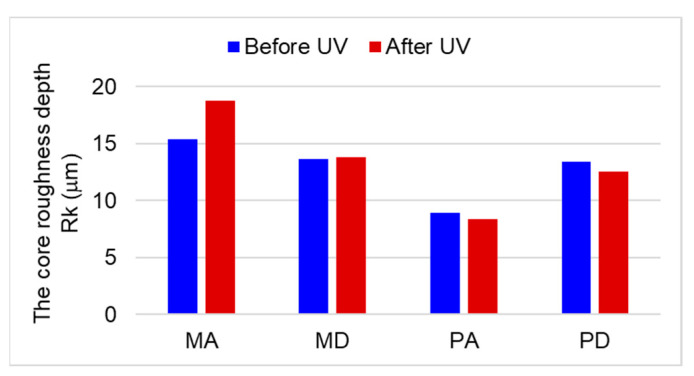
Quantitative analysis of the core roughness depth (Rk) before and after UV treatment.

**Figure 8 polymers-15-01794-f008:**
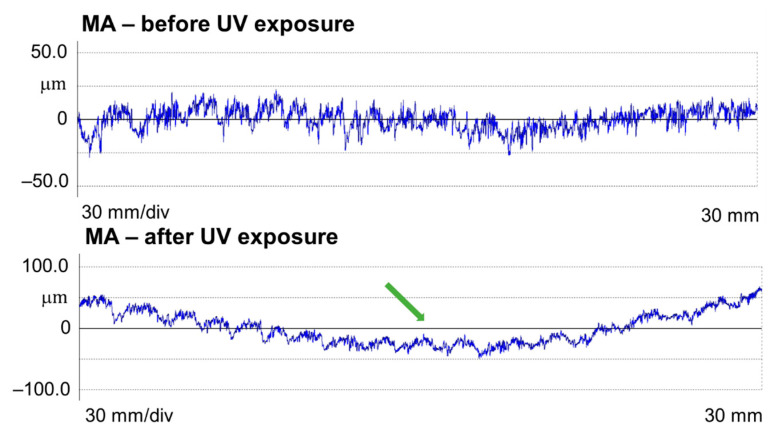
Measured profile of MA spruce samples before and after UV treatment (legend: the deformation/bulging of the piece is marked with the green arrow).

**Figure 9 polymers-15-01794-f009:**
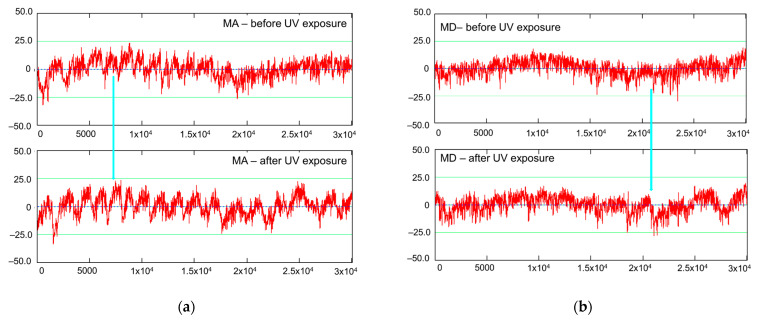
Comparisons of the primary profiles in the case of the MA and MD spruce boards, before (above) and after UV exposure (below): (**a**) sample grade MA; (**b**) sample grade MD (legend: the blue arrow highlights the differences in the roughness profiles).

**Figure 10 polymers-15-01794-f010:**
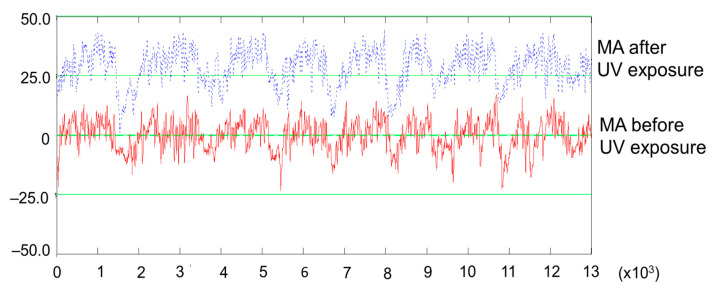
Details of the roughness profiles measured on the MA samples (shown offset and covering a length of 13 mm). Red—before treatment; blue—after UV treatment.

**Figure 11 polymers-15-01794-f011:**
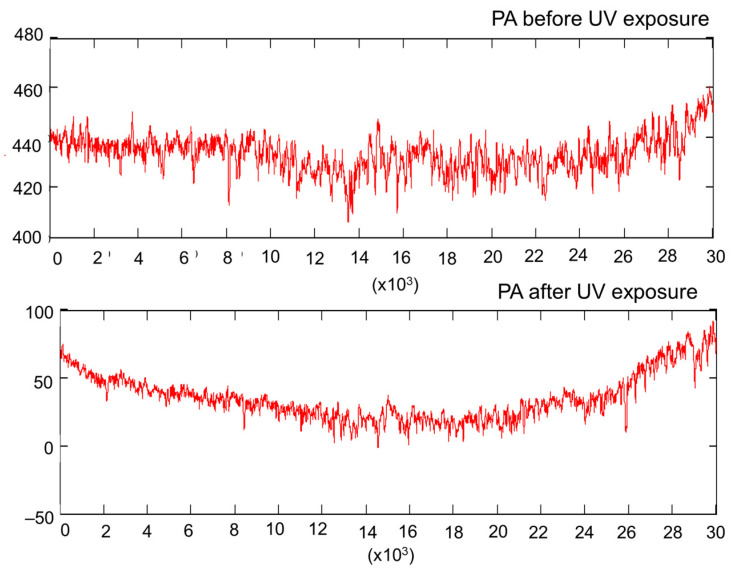
Measured profile of PA sycamore maple samples before and after UV treatment.

**Figure 12 polymers-15-01794-f012:**
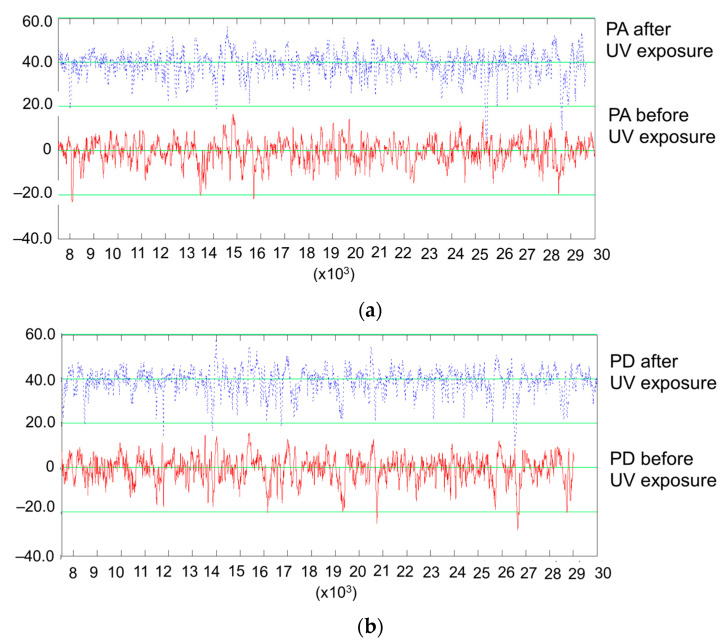
Comparisons of roughness profiles in the case of the PA and PD sycamore maple samples before and after UV exposure: (**a**) sample grade: PA; (**b**) sample grade: PD.

**Figure 13 polymers-15-01794-f013:**
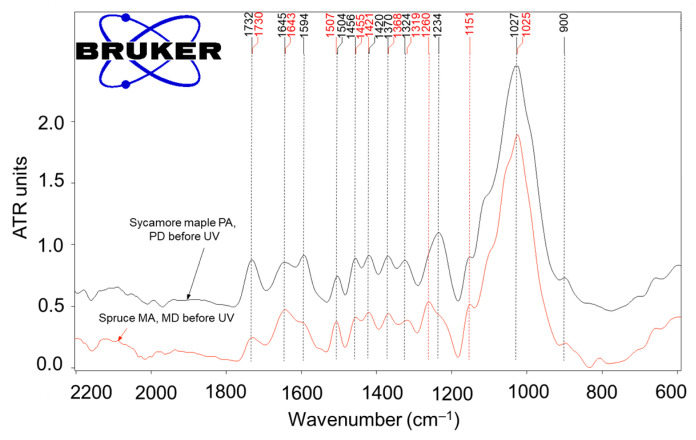
Comparative FTIR spectra for spruce wood and sycamore maple wood samples for the narrow range of 1800–600 cm^−1^ encompassing the fingerprint region.

**Figure 14 polymers-15-01794-f014:**
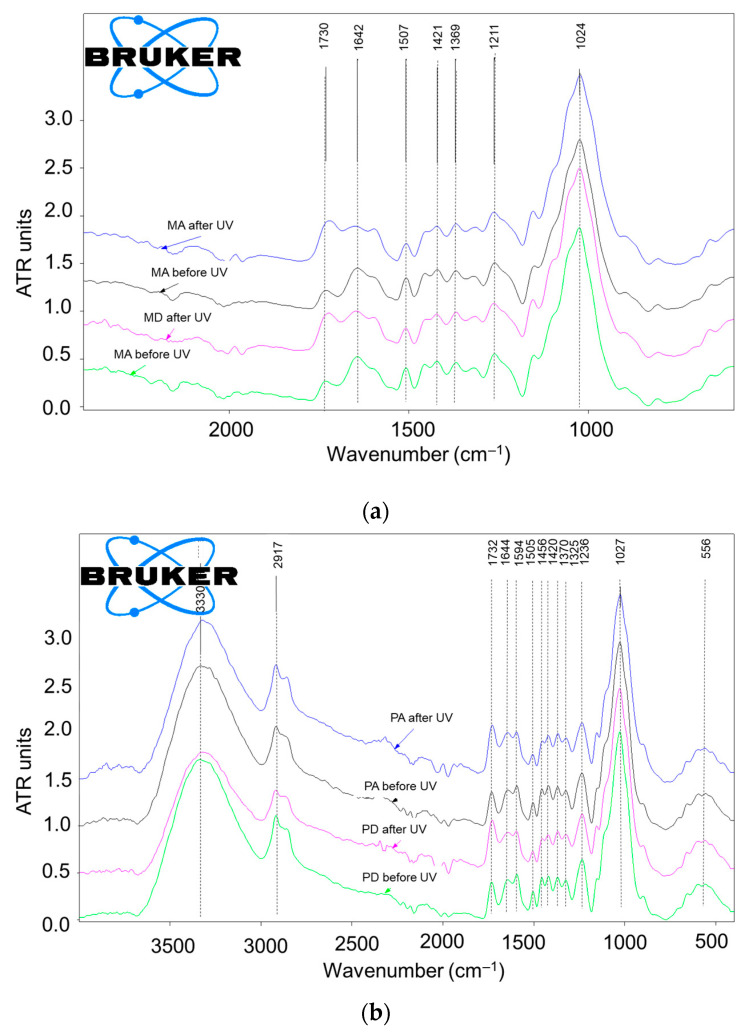
Comparative FTIR spectra (normalized averages) in a narrow range: 2000–600 cm^−1^: (**a**) samples of spruce wood qualities MA and MD before and after UV exposure; (**b**) samples of sycamore maple wood qualities PA and PD before and after UV exposure.

**Figure 15 polymers-15-01794-f015:**
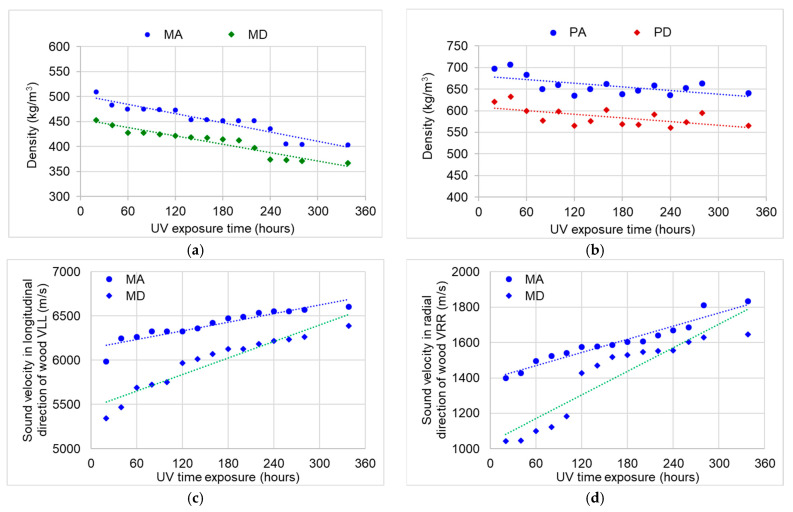
The variation in the physical, acoustic, and elastic parameters with aging duration: (**a**) density variation in the spruce samples; (**b**) density variation in the maple samples; (**c**) variation in sound velocity in spruce wood in the longitudinal direction; (**d**) variation in the sound velocity in the spruce samples in the radial direction; (**e**) variation in the sound velocity in the sycamore maple samples in the longitudinal direction; (**f**) variation if the sound velocity in the sycamore maple samples in the radial direction; (**g**) variation in the Young’s modulus in the spruce samples in the longitudinal direction; (**h**) variation in the Young’s modulus in the spruce samples in the radial direction; (**i**) variation in the Young’s modulus in the sycamore maple samples in the longitudinal direction; (**j**) variation in the Young’s modulus in the sycamore maple samples in the radial direction.

**Figure 16 polymers-15-01794-f016:**
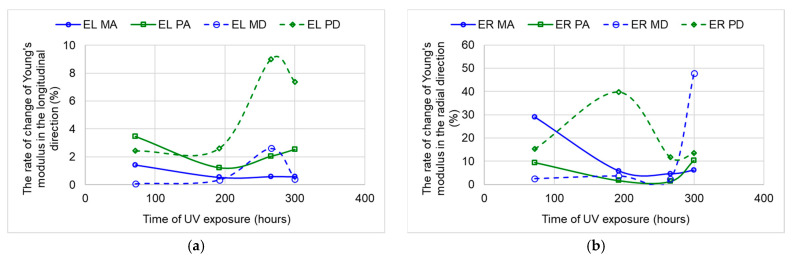
The rate of change in the Young’s modulus with exposure time: (**a**) in the longitudinal direction; (**b**) in the radial direction.

**Table 1 polymers-15-01794-t001:** Color co-ordinates L*, a*, and b*, measured before and after exposure to UV.

Wood Samples		Before UV			After UV			Total Variation		Moisture Content (%)	
		L*	a*	b*	L*	a*	b*	ΔE*	ΔL*	Before UV	After UV
MA	MV	85.63	1.67	20.34	79.12	3.77	32.07	14.75	−7.64	7.73	5.93
	SD	0.21	0.14	0.88	0.13	0.06	0.77			0.25	0.12
MD	MV	86.02	1.67	20.07	79.34	3.67	31.71	15.18	−7.49	7.63	5.57
	SD	1.02	0.47	0.31	0.64	0.22	0.62			0.12	0.4
PA	MV	81.37	3.78	17.23	75.92	4.05	24.71	9.26	−5.45	7.03	4.43
	SD	0.14	0.10	0.30	0.34	0.17	0.48			0.35	0.32
PD	MV	82.03	3.01	16.82	75.68	4.28	25.10	10.50	−6.34	6.60	4.03
	SD	3.35	0.26	0.66	0.98	0.87	0.47			0.85	0.25

**Table 2 polymers-15-01794-t002:** The average values (MV) of the roughness parameters (in microns) and their standard deviations (SD), as measured before and after exposure to UV.

Wood Samples		Ra	Rk	Rpk	Rvk	Rk + Rpk + Rvk	Rt	Wa
MA Before UV	MV	4.61	15.37	4.37	6.02	25.75	41.23	5.61
	SD	0.35	1.24	0.42	1.11	2.12	4.64	1.11
MA After UV	MV	5.62	18.76	4.56	7.28	30.60	48.75	18.85
	SD	0.46	2.31	0.50	1.72	2.51	10.48	3.48
MD Before UV	MV	4.37	13.61	3.20	7.43	24.24	39.53	3.64
	SD	0.44	1.30	0.52	0.51	2.32	1.05	0.19
MD After UV	MV	4.47	13.78	3.45	7.37	24.60	40.03	4.46
	SD	0.43	0.93	0.03	0.94	1.89	3.13	0.12
PA Before UV	MV	3.40	8.93	4.36	8.08	21.37	39.76	0.53
	SD	0.60	1.27	1.58	2.16	4.42	4.57	5.42
PA After UV	MV	3.34	8.40	4.80	9.03	22.23	44.53	4.44
	SD	0.46	1.61	1.89	0.77	3.19	5.24	15.22
PD Before UV	MV	4.69	13.44	4.52	9.71	27.66	49.49	4.72
	SD	0.37	1.34	0.90	0.62	1.40	5.99	6.66
PD After UV	MV	4.48	12.52	4.20	10.57	27.29	45.22	3.36
	SD	0.37	1.08	0.78	0.66	0.95	2.90	13.19

**Table 3 polymers-15-01794-t003:** Chemical differences between spruce and sycamore maple wood before UV exposure.

	The Ratio of the Areas Corresponding to the Wave Number		
Samples	A1730/A1370 Nonconjugated Carbonyl/(Cel + HCel)	A1504-1507/A1370 Lignin/(Cel + HCel)	A1730/A1504-1507 Nonconjugated Carbonyl/Lignin
MA before UV	1.87	1.96	0.95
MA after UV	3.91	1.47	2.65
*MA after UV/MA before*	*2.09*	*0.75*	*2.78*
MD before	1.51	1.99	0.76
MD after UV	3.56	1.38	2.58
*MD after UV /MD before*	*2.36*	*0.69*	*3.41*
PA before	3.91	1.11	3.52
PA after UV	5.01	0.91	5.50
*PA after UV /PA before*	*1.28*	*0.82*	*1.56*
PD before UV	3.93	1.37	2.85
PD after UV	4.67	0.84	5.54
*PD after UV/PD before*	*1.19*	*0.61*	*1.94*

**Table 4 polymers-15-01794-t004:** Summary of acoustic and elastic parameters determined before and during exposure to UV. Legend: ρ—density (kg/m^3^); V_LL_—sound velocity in wood in the longitudinal direction, (m/s); V_RR_—sound velocity in wood in the radial direction, (m/s); E_L_—elasticity modulus in the longitudinal direction (GPa); E_R_—elasticity modulus in the radial direction (GPa); MD—mean value; SD—standard deviation.

		Spruce MA					Spruce MD				
		Initial	72 h	192 h	266 h	300 h	Initial	72 h	192 h	266 h	300 h
ρ (kg*m^−3^)	MV	476	459	444	443	442	431	406	405	404	399
	SD	37	13	35	35	36	30	29	30	28	29
		y = −0.1634x + 479.6					y = −0.0563x + 418.62				
V_LL_ (m/s)	MV	6187	6387	6458	6474	6513	5776	5958	5963	6075	6091
	SD	180	108	117	127	134	374	423	238	303	320
		y = 1.7366x + 6125.9					y = 3.3054x + 5443.7				
V_RR_ (m/s)	MV	1529	1538	1602	1643	1704	1329	1377	1400	1426	1459
	SD	42	113	90	59	113	251	286	263	272	243
		y = 44.6x + 1452.6					y = 30.733x + 1306.1				
E_L_ (GPa)	MV	18.27	18.53	18.63	18.74	18.85	14.50	14.51	14.56	14.94	15.00
	SD	2.33	0.94	2.02	2.15	2.23	2.74	2.92	2.17	2.43	2.48
		y = −0.0016x + 18.86					y = 0.006x + 13.72				
E_R_ (GPa)	MV	0.79	1.02	1.08	1.13	1.20	0.81	0.83	0.86	0.88	1.30
	SD	0.12	0.17	0.09	0.10	0.26	0.32	0.35	0.34	0.34	0.33
		y = 0.0016x + 0.77					y = 0.0027x + 0.39				
		**Sycamore maple (PA)**					**Maple PD**				
ρ (kg*m^−3^)	MV	695	651	649	648	646	618	582	581	578	573
	SD	12	12	12	11	10	16	17	18	16	15
		y = −0.1428x + 681.13					y = −0.1408x + 609.17				
V_LL_ (m/s)	MV	4866	4937	5057	5084	5149	4557	4662	4755	4763	4818
	SD	5.20	80.30	15.59	6.35	45.31	260.29	225.16	221.52	238.53	275.65
		y = 0.6465x + 4605.9					y = 0.6465x + 4605.9				
V_RR_ (m/s)	MV	1947	2115	2141	2130	2249	1506	1641	1701	1737	1831
	SD	70.77	70.38	55.99	65.59	43.97	237.23	249.30	226.70	252.63	283.52
		y = 0.9598x + 1960.4					y = 0.9191x + 1533.9				
E_L_ (GPa)	MV	15.78	16.33	16.53	16.87	17.30	11.94	12.23	12.55	13.68	14.69
	SD	2.92	1.37	0.92	1.48	3.47	0.08	0.16	0.13	0.77	0.30
		y = 0.004x + 15.92					y = 0.0005x + 12.94				
E_R_ (GPa)	MV	2.64	2.89	2.94	2.98	3.29	1.11	1.28	1.79	2.00	2.27
	SD	0.136	0.052	0.108	0.103	0.17	0.14	0.05	0.11	0.10	0.17
		y = 0.0026x + 2.5215					y = 0.0048x + 0.9083				

## Data Availability

https://acadia.unitbv.ro/index.php.

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
