# Peer review of "Aging of Wood for Musical Instruments: Analysis of Changes in Color, Surface Morphology, Chemical, and Physical-Acoustical Properties during UV and Thermal Exposure"

_polymers, 2023, doi:10.3390/polym15071794_

Round 1

Reviewer 1 Report

A very well prepared and studied research.

Author Response

Ref. Polymers-2277019

Aging of resonance wood: analysis of physical and acoustic changes during UV exposure

Current title

Aging of wood for musical instruments: analysis of changes in color, surface morphology, chemical and physical-acoustical properties during UV and thermal exposure

Mariana Domnica Stanciu, Lidia Gurau, Maria Cristina Timar, Camelia Cosereanu and Mihaela Cosnita

Authors' Amendments

First we would like to thank the reviewers for carefully going through the manuscript and providing helpful suggestions for its improvement. Thanks to their constructive comments, we are able to present clearly and better version than the original manuscript. All the comments of the reviewers have been considered. In particular, the following changes have been made according to the reviewers' suggestions, highlighted by yellow color in the manuscript.

Thank you for taking the time to evaluate the article and for the evaluations made!

The title of the article was changed following the recommendation made by one of the reviewers.

Reviewer 2 Report

The topic of wood aging is still interesting and current from the point of view of its long-term research. Accelerating the aging process of wood by UV radiation is a well-known method. Some manufacturers of musical instruments (especially violin masters) practically use UV radiation to accelerate the drying of wood varnish.

1) It is worth considering changing the title, which better describes the essence of the research presented in the article. For example: Aging of wood for musical instruments: analysis of changes in optical, chemical and physico-acoustical properties during UV and thermal exposure.

2) Maple wood cannot be considered as resonance wood, but only as wood for making musical instruments. Resonance wood occurs only in coniferous wood species, in this case it is spruce wood.

3) It is also appropriate to mention the Latin names of the spruce and maple wood species in the article.

4) It is necessary to indicate the moisture content of natural spruce and maple wood during experiments. The moisture content of wood is a fundamental data required when presenting the results of measurements of wooden properties.

5) In Table 4, it would be appropriate to present also the average values of physico-acoustical characteristics such as wood density, sound speed and modulus of elasticity of spruce and maple wood in the given physical units before and after UV. It would be appropriate to indicate the values of the physical and acoustical characteristics as presented in the table of the article you cited: https://www.mdpi.com/2073-4360/14/14/2813

6) It would be appropriate to supplement the graphs of the dependence of EL, ER, r on the time of exposure to UV radiation.

7) It is necessary to check all graphs and correct missing axis descriptions, e.g. in Figure 13. (c-d) the description of the "x" axis is missing.

Author Response

Ref. Polymers-2277019

Aging of resonance wood: analysis of physical and acoustic changes during UV exposure

Current title

Aging of wood for musical instruments: analysis of changes in color, surface morphology, chemical and physical-acoustical properties during UV and thermal exposure

Mariana Domnica Stanciu, Lidia Gurau, Maria Cristina Timar, Camelia Cosereanu and Mihaela Cosnita

Authors' Amendments

 First we would like to thank the reviewers for carefully going through the manuscript and providing helpful suggestions for its improvement. Thanks to their constructive comments, we are able to present clearly and better version than the original manuscript. All the comments of the reviewers have been considered. In particular, the following changes have been made according to the reviewers' suggestions, highlighted by yellow color in the manuscript.

 Response to Reviewer 2:

Thank you very much for your time and suggestions to improve our work!

In the attached file, we answered point by point the recommendations and observations.

Reviewer 3 Report

The paper describes the assessment of the sound quality of two types of musical wood, spruce and maple, before and after ageing by UV exposure.

The Introduction shall be improved for typos. The significant contribution to this field should be indicated clearly in the last paragraph of the introduction. And the motivation and novelty of the work must be added in detail.

Latin names, the second part referring to the wood species, should be written in lowercase (lines 85,86).

Line 89: what does it mean: differed in the frequence and width of annual rings"?

Line 94: why the seasoning period was different for both classes?

Figure 1: wood tissue is very pink? It's not a natural colour. Not of sufficient quality of the photo could be a reason.

Lines 108,109: how was the appropriate UV exposure time selected?

Line 111: why the temperature was equal to 50 degrees? Tonewood is not subject to overheating in such temperatures.

Chapter 2.2.2: where were the colour measurements taken? Was there any scheme for applying the measuring head? At how many points the colour measurements were done?

Line 187: has a valid statistical analysis been carried out?

Figure 4b: why is such a shift of the curves at time equal to 0?

Line 232: when were the samples sanded?

Chapter 3.4: Did the wood have the same moisture content before and after ageing? One of the reasons for the change in mechanical and acoustic parameters may be differences in MC. If the samples bulged after the ageing process, was the measurement of the speed of sound carried out correctly? Was the sound wave dispersing too much?

Table 4: is a dot a decimal separator? Does it mean values in thousands? Physical and acoustic parameters are wrongly determined. The spruce density is too high. Speeds do not correspond to generally known values.

Author Response

Ref. Polymers-2277019

Aging of resonance wood: analysis of physical and acoustic changes during UV exposure

Current title

Aging of wood for musical instruments: analysis of changes in color, surface morphology, chemical and physical-acoustical properties during UV and thermal exposure

Mariana Domnica Stanciu, Lidia Gurau, Maria Cristina Timar, Camelia Cosereanu and Mihaela Cosnita

Authors' Amendments

 First we would like to thank the reviewers for carefully going through the manuscript and providing helpful suggestions for its improvement. Thanks to their constructive comments, we are able to present clearly and better version than the original manuscript. All the comments of the reviewers have been considered. In particular, the following changes have been made according to the reviewers' suggestions, highlighted by yellow color in the manuscript.

 Response to Reviewer 3:

Thank you very much for your time and suggestions to improve our work.

In the attached file, we answered point by point to all recommendations and observations.

Round 2

Reviewer 3 Report

The manuscript is enriched with helpful, well-described schematics that help to understand the problems discussed. Therefore, I do not have any comments on the paper.